# Do women with a previous unintended birth subsequently experience missed opportunities for postpartum family planning counseling? A multilevel mixed effects analysis

Otobo I. Ujah[1,2]*, Jason L. Salemi[2], Rachel B. Rapkin[3], William M. Sappenfield[2], Elen M. Daley[2], Russell S. Kirby[2]

1 Department of Obstetrics and Gynaecology, Federal University of Health Sciences, Otukpo, Nigeria,
2 College of Public Health, University of South Florida, Tampa, Florida, United States of America,
3 University of South Florida, Tampa, Florida, United States of America

* otoboujah@yahoo.com

**Data Availability Statement:** The data underlying the results presented in the study are available

## Abstract

Client-provider communication about family planning (FP) remains an important strategy for preventing unintended pregnancy. Yet, the literature lacks empirical studies examining whether and how women's intendedness of a recent pregnancy may impact subsequent receipt of FP counseling. We investigated whether the intendedness of a recent pregnancy is associated with subsequent missed opportunities (MOs) for FP counseling, taking into account compositional and contextual factors. We performed a secondary analysis using pooled data from the 2016, 2017 and 2018 Performance Monitoring and Accountability 2020 cross-sectional surveys conducted in Nigeria, adjusting for complex design effects. Weighted multilevel logistic regression modeling was used to examine the relationships between pregnancy intention and MOs, overall and at the health facility, using two-level random intercept models. In the analytic sample of women within 24 months postpartum ($N = 6479$), nearly 60% experienced MOs for FP counseling overall and even 45% of those who visited a health facility visit in the past 12 months ($N = 4194$) experienced MOs. In the multivariable models adjusted for individual-/household- and community-level factors, women whose recent birth was either mistimed or unwanted were just as likely to have MOs for FP counseling as their counterparts whose pregnancy was intended ($p > 0.05$). Factors independently associated with a MOs include individual/household level factors such as level of education, exposure to FP media, household wealth index and contextual-level variables (geographic region). While evidence that pregnancy intendedness is associated with MOs for FP counseling remains inconclusive, efforts to mitigate these MOs requires prioritizing women's prior pregnancy intentions as well as equipping healthcare providers with the capacity need to provide high-quality client-centered FP counseling, particularly for women whose recent birth was unintended.

from the PMA website https://www.pmadata.org/
data/request-access-datasets https://datalab.
pmadata.org/dataset/doi%A1034976f2wp-7f02
https://datalab.pmadata.org/dataset/doi%
A103497665q3-qk87 https://datalab.pmadata.org/
dataset/doi%A1034976b5eg-a867.

**Funding:** The authors received no specific funding
for this work.

## Introduction

Access to family planning (FP) and contraceptive services, including the provision of contraceptive counseling, remains a cost-effective strategy for avoiding rapid repeat, and unintended pregnancy and ultimately reducing maternal morbidity and mortality [1–4]. Providing access to comprehensive contraceptive information and options ensures women and couples are better able to enact their reproductive agency and effectively make informed choices regarding contraceptive use and the specific method [5]. Thus, effective interpersonal communication between healthcare providers and clients remains key to promoting contraceptive initiation, compliance, continuation, and method switching [6–8]. While there is currently no universally accepted definition of FP and contraceptive counseling, it essentially involves an interactive process between clients and healthcare providers, with a focus on client-centered care that includes information on contraceptive options, accurate usage and understanding of contraceptive side effects while at the same time addressing concerns and respecting the client's preferences [7,9–11].

Contraceptive counseling is an effective behavior change intervention for preventing unintended pregnancies with several advantages. It offers opportunities for discussing the decision to use contraceptives, accessing them, and receiving education about various methods and their usage [10]. Furthermore, it enhances knowledge regarding the risks, benefits, and side effects of contraceptives [12], while allowing women and couples to engage in conversations with healthcare providers about their desired family size, pregnancy spacing, dispelling misconceptions about contraceptives, and facilitating effective communication within couples. Additionally, contraceptive counseling ensures that contraceptive choices are tailored to meet the specific needs and circumstances of women [6,13]. Ultimately, these benefits contribute to increased client satisfaction with contraceptive decision-making and the quality of care received [10].

By facilitating women's ability to choose contraceptive methods that align with their goals, preferences, and accurate use of the chosen method, contraceptive counseling also plays an important role in reducing rates of unintended pregnancy at the population level and in the long term, contributes to fertility declines and alleviation of poverty [10,14–16]. This is particularly important in Nigeria, as in most countries in Sub-Saharan Africa (SSA), where the burden of unintended pregnancy and short interpregnancy intervals is high. Between 2015 to 2019, approximately 121 million unintended pregnancies occurred worldwide each year, accounting for 48% of all pregnancies and resulting in around 64 unintended pregnancies per 1000 women aged 15–49 [17]. Sub-Saharan Africa had the highest rates (91 unintended pregnancies per 1000 women) compared to North America and Europe (43 per 1000) [17].

Evidence suggests that unintended pregnancies have negative implications for health, social well-being, and economic development [18,19]. In SSA, where resource distribution is inequitable, the majority of unintended pregnancies end in unsafe abortions, a leading cause of maternal morbidity and mortality [17,20]. A study by Kibira and colleagues revealed that women often receive insufficient education regarding the potential side effects of contraceptive methods, making it challenging for many women and couples to make informed decisions regarding their contraceptive and reproductive desires [21].

The World Health Organization (WHO) emphasizes the need for integrating FP counseling into the continuum of care for maternal health services, spanning antenatal care, labor and delivery, postnatal care, and well-child clinics [22]. Although contraceptive counseling and contraceptive use are intricately related [10,23], many women who access healthcare services do not receive adequate FP counseling, resulting in low contraceptive use [12,24]. In a recent study, Thiongo and colleagues revealed that approximately 53% of women who were 12–23

months post-delivery and 48% of women 0–11 months post-delivery missed the opportunity for FP counseling, either at the health facility or in the community [25]. Similarly, another study conducted in Ethiopia found that less than one-half (48%) of women accessing health services in the past 12 months received contraceptive counseling [13]. Several factors contribute to limited access and provision of FP counseling, including limited knowledge and training, a reliance on clients to initiate discussions, limited time, and misconceptions about women's pregnancy risk [14].

Several studies have investigated contraceptive counseling in specific populations, such as among people living with HIV, women seeking abortion services and postpartum women [5,13,24–29]. However, empirical research which specifically explores the relationship between a recent unintended birth and the likelihood of receiving contraceptive counseling is lacking. This knowledge gap precludes effort and opportunity to improve person-centered contraceptive care and service delivery, especially for women at risk of repeat unintended pregnancy and short interpregnancy interval. There is, therefore, an urgent need to close this gap by generating evidence regarding factors influencing this relationship which can inform the design and implementation of new and existing contraceptive programs and policies. Furthermore, identifying women at risk of experiencing missing opportunities (MOs) for FP counseling will improve the effectiveness of FP counseling during interactions with maternal and child health (MCH) providers.

In this study, we used nationally representative survey data to empirically examine the influence of the social context and individual/household factors on the relationship between the intendedness of women's recent birth and the likelihood of experiencing MOs for postpartum FP counseling among women of reproductive age (15–49 years) in Nigeria, using a multilevel approach. The aim of this study, among women of reproductive age, was threefold. First, we aimed to identify key social-ecological factors associated with MOs for postpartum FP counseling. Second, we investigated the extent to which previous pregnancy intention is linked to the likelihood of experiencing MOs for postpartum FP counseling. Third, we aimed to determine whether there is between-community variation in the association between pregnancy intention and subsequent MOs for postpartum FP counseling.

## Theoretical frameworks

Behavior theories aid in identifying and explaining the mechanisms between factors associated with health behaviors. In this study, we draw on two theoretical models/frameworks. Firstly, we utilized the Traits-Desire-Intention-Behavior (TDIB) framework, which outlines a series of motivational steps aimed at comprehending human reproductive behavior. This framework encompasses a sequence of psychological processes involving motivational dispositions and conscious states [30]. This framework posits that attributes or traits act as facilitators or barriers to individuals engaging in specific behaviors, which are activated through conscious desires for or against the behavior. These desires are then transformed into intentions through a decision-making process. Subsequently, these intentions are translated into behaviors that result in the attainment or avoidance of particular outcomes. Given the strong correlation between pregnancy intention and contraceptive use [31], the TDIB sequence provides compelling framework towards understanding the relationship between pregnancy desires and the receipt of FP counseling.

To enhance our understanding of the multilevel influences on the relationship between pregnancy intentions and the nonreceipt of FP counseling, we also incorporated the constructs of the social ecological model. Implicitly, this model recognizes that FP counseling is influenced by multiple factors operating at various levels, including intrapersonal, interpersonal,

community, and policy factors within the context of a specific behavior [32,33]. By adopting the social ecological model, we examine the interplay of intrapersonal, interpersonal and contextual level variables on individual-level outcomes through a hierarchical structure. The social ecological model has been adapted in several studies investigating contraceptive behavior [2,31]. By considering the broader social and environmental factors alongside individual characteristics, we gain insights regarding the complex interrelationships among factors associated with the provision of contraceptive and FP counseling services.

## Methods

### Study design, data source and study population

In this study, we analyzed data from the Performance Monitoring and Accountability 2020 (PMA2020) surveys in Nigeria. The PMA2020 survey is a cross-sectional, nationally representative, population-based survey which collects data annually or semi-annually on FP and other reproductive health indicators from households, females aged 15–49 years, and service delivery points (SDPs). The PMA2020 uses smartphone-assisted technology to gather data from selected households. Female enumerators residing within or near the selected enumeration areas (EAs) collected the data using smartphones. Detailed information about the survey methodology has been published elsewhere [34].

Briefly, the Nigeria PMA2020 survey follows a multistage cluster sampling design. In the first stage, states within the geopolitical zones are selected using probability proportional to size (PPS) sampling, with one state chosen from five out of the six geopolitical zones and two from the North-West Zone, which contains a quarter of Nigeria's total population. In the second stage, 302 clusters (enumeration areas) within each state are selected from the National Population Commission's Census master sampling frame from 2006, using PPS [35]. The third stage involves a random selection of 35–40 households per EA [36].

### Ethical considerations

Ethical approval for the survey was obtained from the Johns Hopkins Bloomberg School of Public Health and the National Health Research Ethics Committee of Nigeria. Verbal consent was obtained from women before their participation in the survey. All PMA2020 data are publicly available upon request (https://www.pmadata.org). The protocol for this study was reviewed and designated exempt by the Institutional Review Board at the University of South Florida (USF) (IRB ID: STUDY005438).

### Sample

In our analyses of the effects of pregnancy intention, we used data from a pooled sample of female respondents who participated in the 2016, 2017 and 2018 rounds of the PMA2020 survey. By pooling data from multiple survey rounds, our goal was to ensure a larger sample size and ultimately, increase the statistical power of the current study. During the 2016 survey, 10,131 households (response rate: 97.1%) and 11,054 de facto females (response rate: 97.9%) completed the survey. For the 2017 survey, 10,063 households (response rate: 97.2%) and 11,380 de facto females (response rate: 98.7%) completed the survey, while the 2018 survey had 10,070 households (response rate: 97.5%) and 11,106 de facto females (response rate: 98.1%) completing the survey. We restricted our analytical sample to respondents between the ages of 15 and 49 years who reported not being pregnant and whose most recent childbirth occurred within two years prior to the survey. We excluded women who were already using a contraceptive method after birth, those with missing or invalid data on pregnancy intentions ($n$ = 22) and MOs for FP

counseling ($n$ = 9). After also excluding minimal missing data (0.09%—consisting of 42 cases for survey weights, 10 cases for household wealth index, 9 cases for marital status, 8 cases for exposure to family planning media, 3 cases for parity and 1 case for educational attainment), the final analytical sample comprised 6479 postpartum women, each nested within 877 clusters.

## Measures

**Exposure variable.**   Pregnancy intendedness describes a woman's desire for pregnancy before or at the time of conception [37]. During the survey, pregnancy intention was measured using the conventional timing-based measure of pregnancy intention by asking women to retrospectively recall their preconception pregnancy desires using the following question: "*At the time you became pregnant, did you want to become pregnant then, did you want to wait until later, or did you not want to have any children at all?*". Response options were "*then*", "*later*" and "*not at all*". As with recent studies [38–41], women who answered that their pregnancy was wanted then were classified as having a pregnancy that intended. Conversely, women who indicated a desire for pregnancy later were categorized as having a mistimed pregnancy while women who answered not at all were classified as unwanted.

**Outcome variable.**   Missed opportunities for FP counseling occur when a woman at risk of unintended pregnancy does not receive FP counseling during a healthcare visit. In this study, receipt of FP counseling was measured based on women's self-reports to the following questions: (a) "*In the last 12 months, have you visited a health facility or camp for yourself (or your children?)*", (b) "*Did any staff member at the health facility speak to you about family planning methods?*", and (c) "*In the last 12 months, were you visited by a community health worker who talked to you about family planning?*". Participants could respond with "*yes*", "*no*", or "*no response*".

Women were considered to have received FP counseling if they had visited a health facility within the 12 months preceding the survey and if a staff member at the facility had discussed FP methods with them. Additionally, women were considered to have received FP counseling if they had been visited by a community health worker who discussed FP with them within the same time frame. On the other hand, if women had not visited a health facility within the 12 months preceding the survey and were not visited by a community health worker for FP discussions within the 12 months preceding the survey, they were considered to have a missed opportunity for FP counseling. In this study, MOs for FP counseling was modeled as a dichotomous variable (yes, no). A second measure to capture MOs for postpartum FP counseling at the health facility for postpartum women who visited a health facility in the last 12 months but did not receive any FP counseling was defined and modeled as a dichotomous variable (yes vs no).

**Control variables.**   The selection of covariates for the models was determined through a comprehensive literature review [7,24,42], considering their biological plausibility in the exposure-outcome relationship, and the availability of variables in the survey data that were consistently measured across all PMA2020 survey rounds included in this study. Following the social ecological model [31], these variables were classified into two levels: individual/household and community factors. At the individual/household level, the following factors were considered: the woman's age at the time of the survey (15–24, 25–34, and 35–49 years), highest level of education attained (less than secondary, secondary, higher than secondary), parity (1–2, 3–4, 5 or more children), married or cohabiting (no, yes) and exposed to family planning mass media (no, yes). Exposure to FP mass media was determined based on women reporting exposure to FP messages through at least one media channel, such as radio, television, newspapers, billboards/posters, magazines, brochures/leaflets, or voice/text messages and household wealth index (classified as poorest, poorer, middle, richer, richest).

The community-level demographic variables included place of residence (urban or rural) and region (North Central, North East, North West, South East, South South and South West). Additional community-level variables were generated by aggregating individual-level attributes within the respective clusters. This approach was employed as clusters could be deemed proxies for communities in the PMA2020 survey data, especially considering the absence of direct measurements for community-level characteristics in the survey. The derived variables include the percentage of women with at least a secondary level of education within a community, the percentage of women residing below the middle household income quintile within a community (indicating those in poor households), and the percentage of women exposed to at least one form of FP mass media within a community.

## Statistical analysis

We used SAS version 9.4 to perform all data analyses while figures were generated using R version 4.2.2. All estimates were based on a complete case analysis. Weighted descriptive statistics using the relevant SAS procedures, including means for continuous variables (SURVEYMEANS) and percentages for categorical variables (SURVEYFREQ), are reported to describe the background characteristics of survey respondents. The SAS survey procedures were used to account for the complex survey design. The Rao-Scott Chi-square test was used to examine the differences between the distribution of the covariate with the outcomes.

**Model building strategy.**  We fitted two-level survey-weighted mixed-effect logistic regression models to investigate the association between pregnancy intention and MOs for postpartum FP counseling while adjusting for individual/household- and community-level factors. This approach was employed to account for the hierarchical nature of PMA2020 survey data where women $i$ (Level 1) were nested in communities $j$ (Level 2), and the dichotomous nature of the outcome variables. Doing so, we generated estimates of both the compositional effects (individual/household) and contextual effects (community) on MOs for postpartum FP counseling. The PROC GLIMMIX procedure with a binomial probability distribution and the logit link function was used. All models were estimated using the pseudo-maximum likelihood approach with adaptive quadrature (METHOD = QUAD), which combines adaptive Gauss-Hermite quadrature for numerical integration with Newton-Raphson routines for maximization while the degrees of freedom for the fixed effects were estimated using the CONTAINMENT approximation (DDFM = CONTAIN). Furthermore, Level 1 weights were scaled to minimize bias in the variance parameter estimator [43,44]. For the estimation of Level-2 errors (G-matrix) in the covariance matrix, the TYPE = VC option was used, which assumes a simple structure that estimates separate variances without considering covariances [45].

To test the hypotheses of this study, we first specified a null model, without including Level 1 or Level 2 explanatory variables, only community-specific random effects, to determine the extent of variability in MOs across communities as well as to adjudge the appropriateness of multilevel modeling as a statistical approach to address the research questions for this study. Thereafter, more complex conditional models were fitted by sequentially including the main predictor of interest as well as Level-1 and Level 2 predictors. In the unadjusted conditional model (Model I), the main effect of pregnancy intention on MOs for postpartum FP counseling was estimated; Model II included Model I and adjusted for individual-/household-level variables only while Model III included Model II adjusted for community-level variables. Lastly, a random-slope model was fitted to determine whether the association between pregnancy intendedness and MOs for postpartum FP counseling differed across communities. For

all the models fitted, level 1 (age) and level 2 (community poverty, community literacy and community FP media exposure) continuous predictor variables were grand-mean centered.

Measures of association (fixed effects)

This model is written as

$$\text{Logit}(p_{ij}) = \log[\pi_{ij}/1 - \pi_{ij}] = \beta_0 + \beta_1 X_{1ij} + \ldots + \beta_n X_{nij} + \upsilon_{0j} + \varepsilon_{ij,}$$

Where:

The logit of $p_{ij}$ is denoted as a sum of linear function of the independent variables.

$\pi_{ij}$ is the probability that the $i^{th}$ woman in the $j^{th}$ community had MOs for postpartum FP counseling.

$1 - \pi_{ij}$ is the probability that the $i^{th}$ woman in the $j^{th}$ community did not have MOs for postpartum FP counseling.

$\beta_0$ is the fixed regression intercept associated with MOs for postpartum FP counseling.

$\beta_1, \beta_2, \ldots \beta_n$ are the effect sizes of by the community-level and individual/household-level predictors.

$X_{1ij}, X_{2ij}, \ldots X_{nij}$ are the independent variables defined at the community-level and individual/household-level.

$\upsilon_{0j}$ and $\varepsilon_{ij}$ are at random errors at the community-level and individual-level, respectively. These are assumed to be normally distributed with mean 0 and variance $\sigma_u^2$. These provide an estimate for the variation in MOs for postpartum FP counseling.

**Measures of variation (random effects).** Three standard metrics were used to disaggregate and highlight the magnitude of variability in MOs for postpartum FP counseling across communities, the intraclass correlation coefficient (ICC), Median Odds Ratio (MOR), and Proportional Change in Variance (PCV). The ICC represents the proportion of the total observed individual variability in MOs that is attributable to between-community variability [46]. The within-community variance in logistic regression models is represented by the variance of the standard logistic distribution. By using the logistic distribution variance of approximately 3.29 (or $\pi^2/3$), the ICC is computed as shown below:

$$\text{ICC} = 100*[\tau_{00}/(\tau_{00} + 3.29)], \ \tau_{00} \text{ is the between} - \text{community variance.}$$

The MOR, on the other hand, quantifies the variability between communities by comparing two individuals randomly selected from different communities. In the context of MOs, the MOR describes the median value of the odds ratio between a community at high risk of MOs and a community and low risk of MOs for postpartum FP counseling when two communities are selected at random [47]. It is calculated based on the following equation:

MOR = $\exp[\sqrt{(2 \times \tau_{00})} \times 0.6745]$, where 0.6745 is the $75^{th}$ percentile of the cumulative distribution function of the normal distribution [46,48]. MOR values nearing 1.0 suggest minimal variation among communities, whereas larger MOR values ($\geq 2.0$) indicate a more substantial level of variation between communities [49]. We estimated the proportional change in variance (PCV) to determine the variation explained by the multilevel models. The $\tau_{00}$ value for conditional models (Models I-IV) were compared to that of the null model $[\tau_{00(0)} - \tau_{00(n)}/ \tau_{00(0)}]$. Different models were compared using measures of goodness of fit using the Akaike information criterion (AIC). Models with smaller AIC values indicate better fitting models.

# Results

## Survey and sample characteristics

The data in the final analytical sample for this study comprised postpartum women ($N = 6479$) nested within 877 communities. Overall, the mean (SE) age of the women was 28.1 (0.2) years. Table 1 presents the socioeconomic and demographic characteristics for the overall sample and for women who visited a health facility in the past 12 months. About half of the participants (51.0%) were between the ages 25 and 34 years and approximately 43.1% were of low

**Table 1. Descriptive statistics on individual/household and community characteristics of all, 2016–2018 Nigeria Performance Monitoring for Action 2020 (PMA2020) survey.**

| Background | Overall | | Visited health facility in last 12 months | |
|---|---|---|---|---|
| | Unweighted *N* | Sample distribution Weighted % (95% CI) | Unweighted *N* | Sample distribution Weighted % (95% CI) |
| Year of survey | | | | |
| 2016 | 2189 | 33.4 (29.0–38.1) | 1366 | 32.8 (28.1–37.8) |
| 2017 | 2186 | 34.4 (30.0–39.1) | 1390 | 34.3 (29.6–39.3) |
| 2018 | 2104 | 32.2 (27.7–37.1) | 1440 | 33.0 (28.1–38.3) |
| **Individual/household characteristics** | | | | |
| Age, mean (SE), y | | 28.1 (0.1) | | 28.2 (0.2) |
| Age (years) | | | | |
| 15–24 | 2250 | 31.1 (29.3–32.9) | 1415 | 29.9 (27.8–32.1) |
| 25–34 | 3028 | 48.8 (47.1–50.5) | 2010 | 50.1 (47.9–52.2) |
| 35–49 | 1201 | 20.1 (18.8–21.5) | 769 | 20.0 (18.4–21.7) |
| Parity | | | | |
| Low (1–2) | 2648 | 43.1 (41.3–44.9) | 1774 | 44.9 (42.8–47.2) |
| Medium (3–4) | 1922 | 30.2 (28.7–31.9) | 1257 | 30.6 (28.6–32.6) |
| High (5+) | 1909 | 26.7 (25.1–28.3) | 1163 | 24.5 (22.6–26.5) |
| Educational attainment | | | | |
| Less than secondary | 3911 | 53.7 (49.4–57.9) | 2240 | 47.3 (44.2–50.5) |
| Secondary | 2010 | 34.6 (31.5–37.9) | 1488 | 37.8 (35.4–40.4) |
| Higher than secondary | 558 | 11.7 (10.0–13.7) | 466 | 14.9 (13.1–16.8) |
| Married/in union | | | | |
| No | 320 | 5.9 (5.1–6.8) | 205 | 5.3 (4.3–6.4) |
| Yes | 6159 | 94.1 (93.2–95.0) | 3989 | 94.7 (93.6–95.6) |
| Exposure to family planning media | | | | |
| No | 2406 | 39.6 (36.7–42.5) | 1282 | 33.2 (30.1–36.4) |
| Yes | 4073 | 60.4 (57.5–63.3) | 2912 | 66.8 (63.7–69.9) |
| Household wealth index | | | | |
| Poorest | 2897 | 36.5 (33.2–40.1) | 1592 | 30.9 (27.4–34.5) |
| Poorer | 1613 | 25.3 (22.9–27.9) | 1081 | 25.0 (22.5–27.7) |
| Middle | 847 | 14.8 (13.2–16.6) | 602 | 15.3 (13.5–17.3) |
| Richer | 631 | 12.4 (11.0–13.8) | 512 | 15.2 (13.5–17.1) |
| Richest | 491 | 11.0 (9.6–12.6) | 407 | 13.6 (11.7–15.8) |
| **Community characteristics** | | | | |
| Place of residence | | | | |
| Urban | 2099 | 40.2 (36.2–44.3) | 1586 | 45.6 (41.1–50.2) |
| Rural | 4380 | 59.8 (55.7–63.9) | 2608 | 54.4 (49.8–58.9) |
| Region | | | | |

*(Continued)*

**Table 1.** (Continued)

| Background | Overall | | Visited health facility in last 12 months | |
|---|---|---|---|---|
| | Unweighted *N* | Sample distribution Weighted % (95% CI) | Unweighted *N* | Sample distribution Weighted % (95% CI) |
| North Central | 996 | 15.6 (13.6–17.7) | 652 | 15.5 (13.4–18.0) |
| North East | 647 | 18.3 (16.0–20.8) | 399 | 17.9 (15.1–21.2) |
| North West | 3470 | 34.5 (31.5–37.6) | 2153 | 31.5 (28.0–351) |
| South East | 419 | 7.2 (6.2–8.4) | 298 | 7.8 (6.7–9.2) |
| South South | 406 | 10.0 (8.4–11.7) | 240 | 9.3 (7.8–11.0) |
| South West | 541 | 14.4 (12.8–16.2) | 452 | 17.9 (15.8–20.3) |
| Community literacy, mean (SE)[a], % | 46.3 (1.5) | | 51.4 (1.6) | |
| Community poverty, mean (SE)[b], % | 61.9 (1.5) | | 61.9 (1.5) | |
| Community FP media exposure, mean (SE)[c], % | 60.3 (1.5) | | 64.1 (1.6) | |
| Number of observations (unweighted sample) | 6479 | | 4194 | |
| Population size (weighted sample) | 5848 | | 3904 | |
| No. of primary sampling units (PSUs) | 877 | | 822 | |
| No. of strata | 7 | | 7 | |

Notes: Percentages (%) and 95% confidence intervals (CIs) are weighted to adjusted for the complex design of the survey.

[a] Percentage of women in PSU who have less than a secondary level of educational attainment.

[b] Percentage of women in PSU living in households in the poorest or poorer wealth quintiles.

[c] Percentage of women in PSU who are exposed to at least one form of family planning mass media.

parity with one or two previous births. Approximately 53.4% of women reported having less than a secondary school level of education. Most women were married or in union (94.1%) and were exposed to at least one form of FP mass media (60.4%). About 61.8% of women resided in poor households while 59.8% resided in rural areas. Most participants resided predominantly in the Northern region (68.4%). Across communities, the mean (SE) percentage of women with a secondary or higher level of education was 46.3% (1.5%), ranging between 0% and 100%. Additionally, the mean (SE) percentage of women residing in poor households across communities was 61.9% (1.5%), ranging from 0% to 100%. In addition, across communities, the mean (SE) percentage of women exposed to one form of FP media was 60.3% (1.5%), ranging from 0% to 100%.

Approximately 65% of respondents in the overall sample (*N* = 4194) reported at least one visit to a health facility within the last 12 months. As presented in Table 1, the mean age (SE) was 28.2 (0.2) years. Approximately 50% of women were between the ages 25–34 years while 44.9% were of low parity. Furthermore, 47.3% had less than a secondary level of education. Most women were also married or in union (94.7%), 66.8% of women were exposed to at least one form of FP media while about 56% resided in poor households.

Based on community level characteristics, most women who visited a health facility in the last 12 months resided in rural areas (54.4%) and 64.9% resided in the Northern region. Among women who reported at least one visit to a health facility in the last 12 months the mean (SE) percentage of women with a secondary or higher level of education was 51.4% (1.6%), ranging between 0% and 100% across communities. Additionally, the mean (SE) percentage of women residing in poor households across communities was 56.6% (1.7%), ranging from 0% to 100%. Lastly, across communities, the mean (SE) percentage of women exposed to one form of FP media was 64.1% (1.6%), ranging from 0% to 100%.

### Prevalence of MOs for FP counseling by pregnancy intendedness

Overall, 71% ($N = 4552$) of participants reported their recent pregnancy as intended while 23% ($N = 1559$) and 6.2% ($N = 368$) reported their recent pregnancy as mistimed and wanted, respectively. Nearly 60% experienced a MOs for postpartum FP counseling during health facility and/or CHW visits. In the entire sample, the prevalence of MOs was 57.8%, 59.9% and 60.2% among women who had intended, mistimed and unwanted pregnancy, respectively (Fig 1). Among women who reported a health facility visit in the last 12 months ($N = 4194$), 72%, 22.4% and 5.6% reported their most recent pregnancy as intended, mistimed and unwanted, respectively. Approximately 45.1% ($N = 1967$) of this subsample experienced a MO. Furthermore, as shown in Fig 2, among women who visited a health facility in the past 12 months, 43.6%, 50.8% and 41.5% of women whose pregnancy was intended, mistimed and unwanted, respectively, experienced a MO for FP counseling.

### Multilevel logistic regression

Tables 2 and 3 present the results of the unconditional (null) and conditional models (Models I-III), showing a trend in which AIC values decrease with increasing model complexity, with

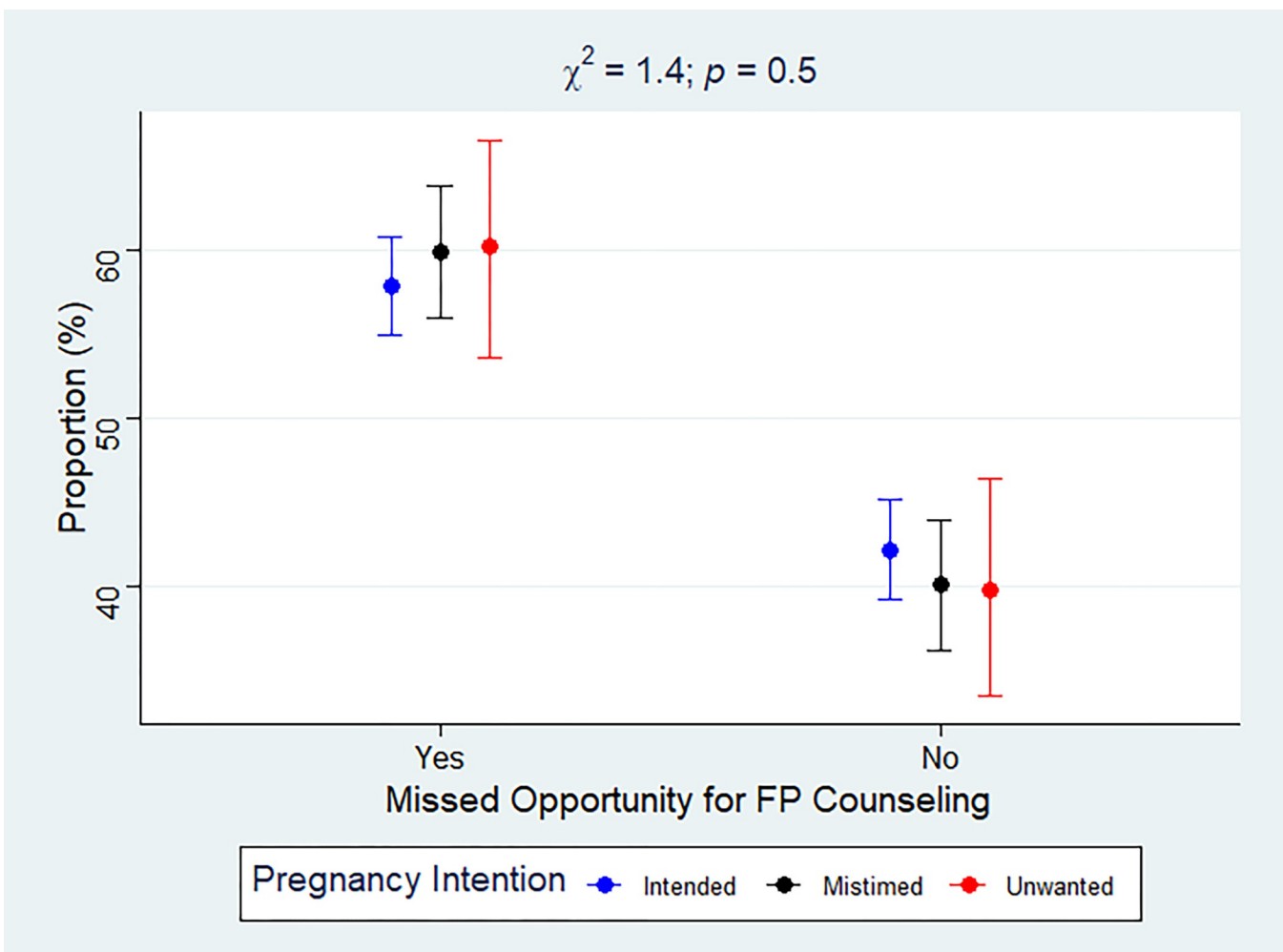

**Fig 1. Women's pregnancy intentions by MO for postpartum FP counseling for the overall sample.** Error bars represent point estimates and 95% confidence intervals for prevalence of MOs.

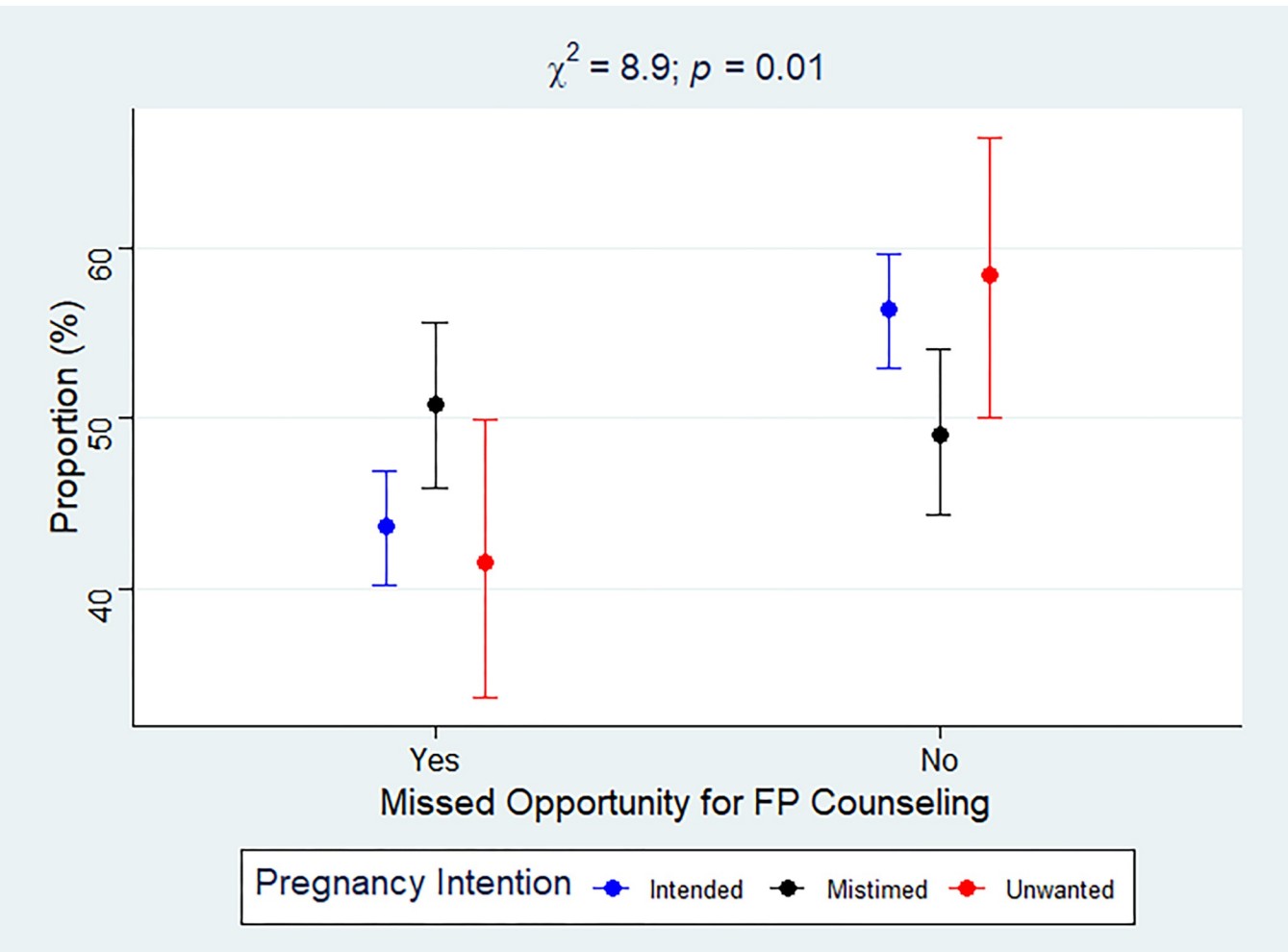

**Fig 2. Women's pregnancy intentions by MO for postpartum FP counseling among those who visited a health facility in the past 12 months.** Error bars represent point estimates and 95% confidence intervals for prevalence of MOs.

Model III indicating better fit since it has the lowest AIC values relative to the other models. Consequently, both the unconditional model and Model III were used to address our research questions for this study.

## Overall MOs for FP counseling

**Unconditional models (null model).** Table 2 presents results from the multilevel models for overall MOs. The estimated intercept for the null model was -0.2, suggesting that in an average community, where the random effect on the logit scale is zero, the odds of experiencing MOs were 0.9, corresponding to a probability of 0.5. This probability, however, varies across communities, as indicated by the statistically significant variability in the likelihood of experiencing a MOs across communities [$\tau_{00} = 4.1$, $z(891.5) = 6.57$, $p < 0.0001$]. Furthermore, the estimated ICC was 0.6, indicating that approximately 60% of the variability observed in MOs can be attributed to systematic differences between the communities in our study. The remaining 40% of the variability is attributable to systematic differences between women or other unknown factors. Furthermore, the MOR was 6.9 indicating that a woman from a

**Table 2. Hierarchical two-level binary logistic regression models of the association between pregnancy intention and MOs for postpartum FP counseling, 2016–2018 Nigeria Performance Monitoring for Action 2020 (PMA2020) survey.** Outcome variable: *Overall (health facility and community health worker) MOs for postpartum FP counseling.*

| Background characteristics | Null model[a] | Model I[b] | Model II[c] | Model III[d] | Model IV[e] |
|---|---|---|---|---|---|
| | OR | OR (95% CI) | OR (95% CI) | OR (95% CI) | OR (95% CI) |
| **Fixed parameters** | | | | | |
| Intercept | 0.9 | 0.9 | **9.2** | **9.0** | **10.5** |
| Year of survey | | | | | |
| 2016 | | | 1.0 | 1.0 | 1.0 |
| 2017 | | | 0.7 (0.4–1.4) | 0.8 (0.4–1.6) | 0.8 (0.4–1.5) |
| 2018 | | | 0.9 (0.6–1.6) | 1.1 (0.7–1.7) | 0.9 (0.6–1.6) |
| **Individual/household** | | | | | |
| Pregnancy intention | | | | | |
| Intended | | 1.0 | 1.0 | 1.0 | 1.0 |
| Mistimed | | 0.9 (0.6–1.2) | 0.9 (0.6–1.2) | 0.9 (0.6–1.2) | 0.9 (0.7–1.4) |
| Unwanted | | 1.1 (0.5–2.7) | 1.0 (0.5–2.3) | 1.0 (0.5–2.4) | 1.0 (0.5–2.3) |
| Age (years)[†] | | | 0.9 (0.9–1.0) | 0.9 (0.9–1.0) | 1.0 (0.9–1.0) |
| Parity | | | | | |
| Low (1–2) | | | 1.0 | 1.0 | 1.0 |
| Medium (3–4) | | | 0.7 (0.4–1.0) | 0.7 (0.5–1.1) | 0.7 (0.4–1.0) |
| High (5+) | | | 0.9 (0.6–1.3) | 0.8 (0.6–1.2) | 0.7 (0.4–1.1) |
| Educational attainment | | | | | |
| Less than secondary | | | 1.0 | 1.0 | 1.0 |
| Secondary | | | **0.3 (0.2–0.7)** | **0.4 (0.2–0.8)** | **0.3 (0.1–0.8)** |
| Higher than secondary | | | **0.3 (0.1–0.7)** | **0.3 (0.1–0.8)** | **0.3 (0.1–0.8)** |
| Married/In union | | | | | |
| No | | | 1.0 | 1.0 | 1.0 |
| Yes | | | 0.6 (0.3–1.1) | 0.6 (0.3–1.0) | 0.6 (0.3–1.1) |
| Exposure to family planning media | | | | | |
| No | | | 1.0 | 1.0 | 1.0 |
| Yes | | | 0.4 (0.3–0.5) | **0.4 (0.3–0.5)** | **0.4 (0.3–0.6)** |
| Household wealth index | | | | | |
| Poorest | | | 1.0 | 1.0 | 1.0 |
| Poorer | | | 1.1 (0.7–1.7) | 1.2 (0.8–1.7) | 1.2 (0.8–1.8) |
| Middle | | | 0.8 (0.5–1.3) | 0.9 (0.7–1.4) | 0.9 (0.7–1.4) |
| Richer | | | **0.6 (0.3–0.9)** | 0.8 (0.5–1.2) | 0.7 (0.4–1.2) |
| Richest | | | 0.7 (0.4–1.6) | 1.1 (0.6–2.2) | 1.1 (0.5–2.2) |
| **Community characteristics** | | | | | |
| Place of residence | | | | | |
| Rural | | | | 1.0 | 1.0 |
| Urban | | | | 0.8 (0.4–1.8) | 0.7 (0.2–1.8) |
| Region | | | | | |
| North Central | | | | 1.0 | 1.0 |
| North East | | | | 1.5 (0.6–3.6) | 1.6 (0.6–4.1) |
| North West | | | | 1.1 (0.4–2.7) | 1.1 (0.4–2.8) |
| South East | | | | 2.0 (0.7–5.7) | 2.1 (0.7–6.2) |
| South South | | | | 1.7 (0.6–4.7) | 1.7 (0.6–4.8) |
| South West | | | | 0.5 (0.2–1.4) | 0.4 (0.1–1.4) |
| Community-level literacy[†] | | | | 0.9 (0.9–1.0) | 0.9 (0.9–1.0) |
| Community-level poverty[†] | | | | 0.9 (0.9–1.0) | 0.9 (0.9–1.0) |

*(Continued)*

**Table 2.** (Continued)

| Background characteristics | Null model[a] | Model I[b] | Model II[c] | Model III[d] | Model IV[e] |
|---|---|---|---|---|---|
| | OR | OR (95% CI) | OR (95% CI) | OR (95% CI) | OR (95% CI) |
| Community-level family planning media exposure[†] | | | | 0.9 (0.9–1.0) | 0.9 (0.9–1.0) |
| **Random parameters** | **Null model** | **Model I** | **Model II** | **Model III** | **Model IV** |
| Community-level variance (SE) | 4.1 (0.6) | 4.2 (0.6) | 3.3 (0.7) | 3.1 (0.7) | 2.9 (0.8) |
| ICC (%) | 55.7 | 55.9 | 49.9 | 48.6 | 46.9 |
| MOR | 6.9 | 7.0 | 5.6 | 5.4 | 5.1 |
| Explained variance (PCV, %) | Reference | -2.4 | 19.5 | 24.4 | 29.3 |
| Community slope variance | | | | | 1.1 (0.4) |
| **Model fit statistics** | | | | | |
| AIC | 8813.8 | 8814.1 | 8273.6 | 8222.4 | 8157.1 |
| **Sample size** | | | | | |
| Community-level | 877 | 877 | 877 | 877 | 877 |
| Individual-level | 6479 | 6479 | 6479 | 6479 | 6479 |

*Notes*: Estimation Method = Pseudo-maximum likelihood; Containment degrees of freedom; Probability distribution = binomial; Link function = logit. Values in bold significant at $p < 0.05$.

[†]Variables were grand-mean centered.

Abbreviations: OR–odds ratio, CI–confidence interval, ICC-Intraclass correlation coefficient, MOR–median odds ratio, PCV–Proportional Change in Variance, AIC–Akaike Information Criteria.

[a] Null model unconditional model, baseline model without any predictor variables.

[b]Model I–includes the main explanatory variable (pregnancy intention).

[c]Model II–Model I adjusted for only individual/household-level characteristics.

[d]Model III–Model I adjusted for individual/household-level and community-level characteristics (full model).

[e]Model IV–Random slopes model adjusted for individual/household-level and community-level characteristics.

community with high MOs had nearly seven-fold higher odds of MOs compared to a woman from a community with low MOs.

**Conditional models with main explanatory variable, Level 1 and Level 2 predictors.** Table 2 shows the results of the overall association between previous pregnancy intention and subsequent MOs. The results from the best fitting model (Table 2, Model III), indicate that, after controlling for individual/household and community level factors, having a recent mistimed pregnancy, as compared to having an intended pregnancy, was associated with a 10% lower odds (aOR = 0.9, 95% CI = 0.6–1.2) of experiencing MOs for postpartum FP counseling. However, this association was not statistically significant. Also, women who had a recent unwanted pregnancy, compared to those whose recent pregnancy was intended, showed no significant difference in the likelihood of experiencing MOs for postpartum FP counseling (aOR = 1.0, 95% CI = 0.5–2.4).

In the fully adjusted model (Table 2, Model III), only individual/household level (educational attainment and exposure to FP media) but not community level variables were significantly associated with MOs for postpartum FP counseling. The odds of experiencing MOs for postpartum FP among women with secondary and higher than secondary level educational attainment were 0.4 (95% CI = 0.2–0.8) and 0.3 (95% CI = 0.1–0.8) times lower, respectively, than for women with a less than secondary level of education. Similarly, relative to those with no exposure to any form of FP media, women exposed to at least one form of FP mass media had a 60% lower odds of experiencing MOs for postpartum FP counseling (OR = 0.4, 95% CI = 0.3–0.5). The PCV of Model III relative to the null model suggests that addition of both the individual/household and community level factors explained 24.4% of the differences in

**Table 3. Hierarchical two-level binary logistic regression models of the association between pregnancy intention and missed opportunity for postpartum FP counseling, 2016–2018 Nigeria Performance Monitoring for Action 2020 (PMA2020) survey.** *Outcome variable*: *Missed opportunity for family planning counseling at a health facility, N = 4194.*

| Background characteristics | Null model[a] | Model I[b] | Model II[c] | Model III[d] | Model IV[e] |
|---|---|---|---|---|---|
| | OR | OR (95% CI) | OR (95% CI) | OR (95% CI) | OR (95% CI) |
| Fixed effects | 0.4 | **0.5** | **4.6** | **5.5** | **7.0** |
| Intercept | | | | | |
| Year of survey | | | | | |
| 2016 | | | 1.0 | 1.0 | 1.0 |
| 2017 | | | 0.6 (0.3–1.2) | 0.6 (0.3–1.4) | 0.6 (0.3–1.3) |
| 2018 | | | 0.7 (0.4–1.3) | 0.8 (0.5–1.4) | 0.7 (0.4–1.3) |
| **Individual/household** | | | | | |
| Pregnancy intention | | | | | |
| Intended | | 1.0 | 1.0 | 1.0 | 1.0 |
| Mistimed | | 1.1 (0.7–1.6) | 0.6 (0.6–1.4) | 0.9 (0.6–1.3) | 1.0 (0.6–1.7) |
| Unwanted | | 0.9 (0.4–2.8) | 1.1 (0.5–2.6) | 1.1 (0.4–2.6) | 0.9 (0.3–2.2) |
| Age (years)[†] | | | 0.9 (0.9–1.0) | 0.9 (0.9–1.1) | 0.9 (0.9–1.1) |
| Parity | | | | | |
| Low (1–2) | | | 1.0 | 1.0 | 1.0 |
| Medium (3–4) | | | 0.7 (0.4–1.1) | 0.7 (0.5–1.1) | 0.7 (0.4–1.1) |
| High (5+) | | | 0.9 (0.5–1.4) | 0.8 (0.4–1.4) | 0.7 (0.4–1.1 |
| Educational attainment | | | | | |
| Less than secondary | | | 1.0 | 1.0 | 1.0 |
| Secondary | | | **0.3 (0.1–0.8)** | **0.3 (0.1–0.8)** | **0.3 (0.1–0.7)** |
| Higher than secondary | | | **0.2 (0.1–0.6)** | **0.2 (0.1–0.6)** | **0.2 (0.1–0.5)** |
| Married/In union | | | | | |
| No | | | 1.0 | 1.0 | 1.0 |
| Yes | | | 0.6 (0.3–1.3) | 0.6 (0.3–1.2) | 0.5 (0.2–1.3) |
| Exposure to family planning media | | | | | |
| No | | | 1.0 | 1.0 | 1.0 |
| Yes | | | **0.4 (0.3–0.6)** | **0.4 (0.3–0.6)** | **0.4 (0.3–0.6)** |
| Household wealth index | | | | | |
| Poorest | | | 1.0 | 1.0 | 1.0 |
| Poorer | | | 0.7 (0.4–1.1) | 0.8 (0.4–1.4) | 0.8 (0.4–1.4) |
| Middle | | | 1.0 (0.6–1.9) | **1.6 (1.0–2.5)** | **1.6 (1.0–2.6)** |
| Richer | | | 0.7 (0.4–1.3) | 1.3 (0.6–2.6) | 1.2 (0.6–2.7) |
| Richest | | | 0.9 (0.4–2.2) | 2.0 (0.8–5.0) | 1.9 (0.8–5.2) |
| **Community characteristics** | | | | | |
| Place of residence | | | | | |
| Rural | | | | 1.0 | 1.0 |
| Urban | | | | 0.6 (0.3–1.4) | 0.5 (0.2–1.3) |
| Region | | | | | |
| North Central | | | | 1.0 | 1.0 |
| North East | | | | **2.9 (1.1–8.1)** | **3.7 (1.3–10.4)** |
| North West | | | | 1.2 (0.5–3.2) | 1.3 (0.5–3.2) |
| South East | | | | 1.7 (0.5–5.2) | 1.8 (0.5–5.7) |
| South South | | | | 0.9 (0.3–2.9) | 0.9 (0.3–2.9) |
| South West | | | | 0.3 (0.1–1.1) | 0.3 (0.1–1.1) |
| Community-level literacy[†] | | | | 1.0 (0.9–1.0) | 1.0 (0.9–1.0) |
| Community-level poverty[†] | | | | 1.0 (0.9–1.0) | 1.0 (0.9–1.0) |

*(Continued)*

**Table 3.** (Continued)

| Background characteristics | Null model[a] | Model I[b] | Model II[c] | Model III[d] | Model IV[e] |
|---|---|---|---|---|---|
| | OR | OR (95% CI) | OR (95% CI) | OR (95% CI) | OR (95% CI) |
| Community-level family planning media exposure[†] | | | | 0.9 (0.9–1.0) | 0.9 (0.-1.0) |
| **Random effects** | **Null model** | **Model I** | **Model II** | **Model III** | **Model IV** |
| Community-level variance (SE) | 4.9 (0.8) | 4.8 (0.8) | 4.4 (0.8) | 3.9 (0.9) | 3.5 (0.9) |
| ICC (%) | 60.2 | 59.4 | 56.9 | 54.7 | 51.1 |
| Explained variance (PCV, %) | Reference | 3.5 | 12.9 | 20.4 | 31.0 |
| MOR | 8.4 | 8.0 | 7.3 | 6.7 | 5.8 |
| Community slope variance | | | | | 1.7 (0.6) |
| **Model fit statistics** | | | | | |
| AIC | 5867.5 | 5825.5 | 5549.8 | 5470.7 | 5412.33 |
| **Sample size** | | | | | |
| Community-level | 822 | 822 | 822 | 822 | 822 |
| Individual-level | 4194 | 4194 | 4194 | 4194 | 4194 |

*Notes*: Estimation Method = Pseudo-maximum likelihood; Containment degrees of freedom; Probability distribution = binomial; Link function = logit. Values in bold significant at $p < 0.05$.

[†]Variables were grand-mean centered.

Abbreviations: OR–odds ratio, CI–confidence interval, ICC-Intraclass correlation coefficient, MOR–median odds ratio, PCV–Proportional Change in Variance, AIC–Akaike Information Criteria.

[a] Null model unconditional model, baseline model without any predictor variables.

[b]Model I–includes the main explanatory variable (pregnancy intention).

[c]Model II–Model I adjusted for only individual/household-level characteristics.

[d]Model III–Model I adjusted for individual/household-level and community-level characteristics (full model).

[e]Model IV–Random slopes model adjusted for individual/household-level and community-level characteristics.

overall MOs for postpartum FP counseling. Furthermore, the results of the random slopes model, which aimed to determine if the effects of pregnancy intention on subsequent overall MOs varied across different communities, demonstrated no significant differences in these effects across communities.

## Missed opportunity for postpartum FP counseling at the health facility

**Unconditional models (null models).** The results of the multilevel models for MO for FP counseling during a health facility visit are shown in Table 3 The estimated intercept for the null model was -0.9 indicating that in a typical average community, the odds of missing an opportunity to receive FP counseling at the health facility was 0.4. This corresponds to a probability of 0.3. The output from the null model revealed a statistically significant variability in the log odds of experiencing a MO for FP counseling at the health facility between communities [$\tau_{00}$ = 4.9, z(845.2) = 5.9, $p < 0.0001$]. The ICC for experiencing a MO for FP counseling at the health facility was 0.6, suggesting that approximately 60% of the variability in MO for FP counseling at the health facility is accounted for by the systematic differences by communities in our study, leaving 40% of this variability to be accounted for by systematic differences between women or other unknown factors. The MOR was 6.08. The change in variability from Model III (full model) demonstrates that the 32.0% reduction in the odds of MO for FP counseling at the health facility can be accounted for by individual-/household-level and community-level characteristics. The MOR was 8.4 indicating that a woman from a community with high MO for FP counseling at the health facility had 8.4 times the odds of MO for FP

counseling at the health facility compared to a woman from a community with low MO for FP counseling, suggesting that the between-community variation for MO for FP counseling at the health facility was high.

**Conditional models with main explanatory variable, Level 1 and Level 2 predictors.**
Table 3 shows the result of the multilevel analyses estimating the association between pregnancy intention and subsequent MO for postpartum FP counseling at the health facility. The results from the best fitting model (Table 3, Model III), indicate that, after controlling for individual/household and community level factors, having a recent mistimed pregnancy, as compared to having an intended pregnancy, was associated with a 10% lower odds (aOR = 0.9, 95% CI = 0.6–1.3) of experiencing MO for postpartum FP counseling at the health facility. However, this association was not statistically significant. In contrast, women who had a recent unwanted pregnancy, compared to those whose recent pregnancy was intended, had a 10% higher odds of experiencing MO for postpartum FP counseling at the health facility (aOR = 1.1, 95% CI = 0.4–2.6). This association, nevertheless, was not statistically significant.

In the model adjusted for individual/household and community level factors (Table 3, Model III), several individual/household level (educational attainment, exposure to FP media and household wealth index) and community level (geographic region) variables were significantly associated with experiencing MO for postpartum FP counseling at the health facility. The odds of experiencing MO for postpartum FP counseling at a health facility among women with secondary and higher than secondary level educational attainment were 0.3 (95% CI = 0.1–0.8) and 0.2 (95% CI = 0.1–0.6) times lower, respectively, than for women with a less than secondary level of education. Also, relative to those with no exposure to any form of FP media, women exposed to at least one form of FP mass media had a 60% experiencing MO for postpartum FP counseling at a health facility (aOR = 0.4, 95% CI = 0.3–0.6). Women residing in households with an average wealth index had approximately 1.6 times the odds of experiencing a MO for postpartum FP counseling at a health facility compared with women residing in households with the poorest wealth index (aOR = 1.6, 95% CI = 1.0–2.5).

At the community-level, after controlling for individual/household-level factors, women residing in the North East region had a nearly 3-fold higher odds of experiencing a MO for postpartum FP counseling at a health facility compared with women residing in the North Central region (aOR = 2.9, 95% = 1.1–8.1)

The PCV of Model III relative to the null model suggests that addition of both the individual/household and community level factors explained 20.4% of the differences in MO for postpartum FP counseling at a health facility. Furthermore, the results of the random slopes model, showing whether the effects of pregnancy intention on subsequent MO for postpartum FP counseling at a health facility varied across different communities, did not reveal significant differences in these effects across communities.

## Discussion

In this study, we employed a multilevel approach to examine the association between the intendedness of women's most recent pregnancy (in the past two years) and the likelihood of experiencing MOs for postpartum FP counseling, using three population-based nationally representative cross-sectional rounds of PMA2020 survey data collected in Nigeria. We found that nearly 60% of women experienced MOs for postpartum FP counselling either at the health facility or in the community, while 45% of postpartum women who had visited a health facility in the past 12 months experienced MOs. These estimates are slightly higher but relatively comparable with the findings reported in the study by Thiongo and colleagues, where they found an overall MO rate of 50.4% and a health facility MO rate of 41.8% [25]. Another study in

Tanzania showed that about 60% of women reported that they did not receive FP counseling during a visit to any health facility [5]. We observed that the prevalence of MOs in our study was higher among women with unintended pregnancy. These findings may suggest that client-provider communication about FP, especially among postpartum women who experienced an unintended pregnancy, might be occurring less frequently during healthcare contact (s). The magnitude of MOs in our study also supports empirical evidence which suggests that unmet need for FP services, including contraceptive counseling, is highest in the first 24 months postpartum [50]. Therefore, efforts to expand access to, and coverage of postpartum FP need to also take into consideration improvement in client-provider communication about FP during this critical window, especially for women at risk of rapid repeat and, unintended pregnancy.

Overall, our results suggest that after accounting for individual/household and community-level factors, women whose recent pregnancy was mistimed or unwanted were just as likely to experience MOs for FP counseling at the health facility and/or during CHW visits as their counterparts whose pregnancy was intended. This finding is consistent with results from prior research which also did not reveal any significant association between pregnancy intendedness and MOs for FP counseling [25]. There are several plausible explanations for this lack of an observed effect. First, the proportion of MOs for FP counseling was generally balanced among women in the sample, and therefore, not distinctive for women based on the intendedness of their recent pregnancy. This may suggest that postpartum women in Nigeria, regardless of their previous pregnancy intentions, were generally less likely to receive FP counseling. While MOs for a healthcare service among eligible individuals are mainly related to provider attitudes and behaviors [51,52], this finding is indeed surprising, given the assumption that women who previously experienced an unintended pregnancy would have strong motivations for preventing future unintended pregnancies and therefore likely to seek guidance on the adoption of effective contraceptive methods. The lack of a significant difference as shown in our study, however, challenges this assumption and suggests the possibility of a complex interplay of multiple factors, warranting further investigation.

Second, our operational definition of missed opportunities (MOs) for FP counseling was indirectly established based on women's self-report of receiving FP counseling either in the health facility, community, or both. Consequently, our determination of a 'missed opportunity' was contingent upon our subjective interpretation rather than direct reporting by the female survey respondent. This approach poses substantial methodological concerns. First, is the potential for recall bias regarding women's receipt of FP counseling, which may have biased our results towards the null, potentially masking the true relationships between pregnancy intendedness and subsequent MOs for FP counseling. The other concern is that our operationalization of MOs relied based on health facility and/or CHW visits raises concerns regarding whether existing approach(es) for ascertaining MOs for FP counseling indeed capture what it is intended to measure, considering the possibility of women receiving FP counseling from alternative sources outside of these settings. This underscores the need for more appropriate, validated and standardized approaches which will better capture MOs and facilitate comparisons of findings across studies and different contexts.

Based on the model that best fits our data, we found that only individual factors such as level of educational attainment and exposure to FP media were associated with MOs for FP counseling during health facility and/or CHW visits. However, among women who visited a health facility within the past 12 months, in addition to the aforementioned individual-level factors, household wealth index and geographic region were identified as household and community-level factors, respectively, which were associated with MOs. This may suggest that

women's general and contraceptive literacy as well as social economic status may influence patient-provider communication about FP.

We also found that communities accounted for a significant proportion of the variation between women in MOs. Our results suggest that the community in which women resided accounted for an initial 55%-60% of the variation in MOs among women in our sample. However, this community-level variation decreased substantially with the inclusion of individual/ household and community level factors in the model and therefore provide evidence that factors at both levels of influence contribute substantially to the geographic distribution of MOs. In addition, we found no evidence that the effect of pregnancy intention on MOs for postpartum FP counseling did not differ across communities, implying that pregnancy intention was not more important for MOs in some communities relative to others. Furthermore, our results demonstrate that, if a postpartum woman moved to another community with a higher probability of MOs, the odds of experiencing MOs, facility are likely to each increase by at least five-fold. Therefore efforts to strengthen FP policies and programs should consider not only individuals at higher risk of MOs but also high-risk communities as well.

Despite the observed null effect of pregnancy intention on MOs, our study makes substantial contributions to existing literature by highlighting the magnitude of unmet need for postpartum FP counseling and associated factors, overall and especially for women whose pregnancy was unintended. Furthermore, it provides additional insight into existing gaps regarding client-provider communications about FP in Nigeria by suggesting that providers may not be routinely assessing the intendedness of women's recent pregnancy when providing FP counseling. Such gaps may further contribute to inconsistent, inaccurate or the nonuse of contraception and ultimately result in future unintended pregnancy [53]. In addition, the contextual-level findings of our study may reflect the prevailing normative climate in Nigeria which generally does not prioritize FP usage. As a result, providers are likely to overlook the need to provide information on FP to postpartum women or may be less likely to adhere MCH protocols and guidelines to screen for past unplanned pregnancies. On the other hand also, postpartum women may be reluctant to inquire about options for delaying or avoiding pregnancy in public health facility settings. Therefore, there is a need for future research to explore how and to what extent community attitudes and social norms regarding FP influence providers' willingness to offer FP information and postpartum women's attitudes and comfort with receiving such information after delivery.

As much remains to be understood regarding the socio-ecological factors associated with MOs for FP counseling and the variations that exist across communities, the results of our study can inform the design of future surveys and research to generate additional evidence delineating the mechanisms which underpin healthcare provider behaviors towards FP counseling. The high prevalence of MOs observed in our study may also reflect challenges associated with integrating quality FP counseling into other points of care. Evidence suggests that provision of FP counseling at health facilities may vary depending on the services provided, the available health workforce, time constraints, patient numbers and an enabling environment for counseling that considers client privacy [54]. Overcoming these health systems barriers may also be critical to strategy to mitigate MOs for FP services.

Notably, the results of our study, though weak, partially support the applicability of the TDIB framework in understanding women's non-receipt of FP counseling, based on results of the best fitting model, which demonstrated only modest success in predicting women who were less likely to receive FP counseling during health and/or CHW visits. This could partly be explained by the fact that our study was based on a secondary analysis of data which was not primarily designed for the purpose of testing the TDIB framework. Furthermore, the framework was missing on intention as a construct. However, the inclusion of fertility intentions in

the model, along with pregnancy intendedness would pose important concerns given the likelihood of a strong correlation between both variables. Nonetheless, the theoretical approach followed in our study provides evidence for future studies to leverage in order to develop more specific measures of the TDIB constructs which ultimately could potentially improve model fit and provide explanations regarding the mechanisms underpinning women's receipt of FP counseling. Additionally, although the general contextual effects (ICC, MOR and PCV) indicate that communities may explain greater amounts of variation in the receipt of FP counseling, we found inconclusive evidence that specific contextual effects or variables such as place of residence, community-level poverty and community-level literacy on the other hand may do so. These findings highlight the limitations social context may play in the ability of the TDIB framework to explain women's receipt of FP counseling.

By drawing from the social-ecological model and also employing a multilevel analytic strategy to examine MOs, our study not only investigated individual and household level factors influencing MOs, but also provided insight into the overall contextual effect. By doing so, we quantified the variability in MOs between communities. Furthermore, the use of data from nationally representative samples enhances the likelihood that the findings and conclusions of this study accurately reflect patterns and relationships pertaining to MOs within the larger population, which otherwise would have been obscured in smaller, non-representative samples. Most research on pregnancy intention often operationalize this construct dichotomously, categorized as either unintended or intended. Such approach has methodological limitations as it does not capture the differential effects of unintended pregnancy [55]. Our study overcomes this limitation by adopting a more nuanced approach, operationalizing pregnancy intention as a polytomous variable.

Our study also has several limitations. First, although we adjusted for potential multilevel confounders, however the possibility of residual confounding due unmeasured provider-specific and health care system-specific factors contributing to MOs could have influenced our results and the extent to which our conclusions are valid. In addition, our study focused on postpartum women who had given birth within the past 24 months and had received FP counseling within the last 12 months. As the PMA data collection does not capture information about whether these women had received FP counseling in previous years, it is likely that women may have received postpartum FP counseling at some point but not in the past year, thus limiting the interpretation and generalizability of the findings. Lastly, the cross-sectional nature of the data used in this analysis precludes our ability to make inferences about a temporal and causal relation between women's previous birth intendedness and postpartum MOs for FP counseling.

Future population-based surveys should be modified to capture provider- and health-system-related factors related to contraceptive counseling, as doing so may strengthen the validity of study results. Furthermore, advancements in technology provide an opportunity for facility-based studies to leverage the use of electronic health records linking patient and provider databases which can overcome challenges with bias in measuring pregnancy intentions and receipt of FP counseling. Also, studies employing exploratory or qualitative approaches can comprehensive perspectives regarding the quality of client-provider conversations that either enable or prevent opportunities for postpartum contraceptive adoption.

As health provider recommendations remain an important strategy to improving contraceptive utilization rates [56,57], there is a need to strengthen the capacity of healthcare providers at various levels of the healthcare system to provide quality FP information to clients. Equally important is the need to evaluate the readiness, availability and expertise of health facilities and providers in the provision of FP counseling. Furthermore, guidance and practice recommendations on FP should include and emphasize the need for providers to assess the

intendedness of women's recent pregnancies. This will ensure that FP counseling is tailored to address the specific needs of those at risk of rapid repeat pregnancies and future unintended pregnancies. The better equipped providers are, the more likely they will be to provide women with the services that will enable them to make informed FP choices.

## Conclusion

The findings from this study suggest that although MOs was high among postpartum women in Nigeria, there was insufficient evidence to conclude that unintended pregnancy was associated with MOs, overall or at the health facility. As MOs for postpartum FP counseling represents a missed opportunity to prevent unintended pregnancy, there is need to explore innovative, multilevel strategies to strengthen FP policies and programs. These efforts should prioritize screening postpartum women based on the intendedness of their recent pregnancy, provision of high-quality FP counseling, particularly targeting women who have experienced unintended pregnancies. Such efforts hold the potential to significantly mitigate MOs and ultimately reduce both incident and recurrent unintended pregnancies.

## Acknowledgments

The authors would like to thank the reviewers of this manuscript for their valuable and insightful comments which substantially improved the rigor, quality and readability of this manuscript.

## Author Contributions

**Conceptualization:** Otobo I. Ujah, Jason L. Salemi, Rachel B. Rapkin, William M. Sappenfield, Elen M. Daley.

**Data curation:** Otobo I. Ujah.

**Formal analysis:** Otobo I. Ujah, Jason L. Salemi, Russell S. Kirby.

**Investigation:** Otobo I. Ujah.

**Methodology:** Otobo I. Ujah, Jason L. Salemi, Rachel B. Rapkin, William M. Sappenfield, Elen M. Daley, Russell S. Kirby.

**Software:** Otobo I. Ujah.

**Supervision:** Jason L. Salemi, Rachel B. Rapkin, William M. Sappenfield, Elen M. Daley.

**Validation:** Jason L. Salemi, Rachel B. Rapkin, William M. Sappenfield, Elen M. Daley, Russell S. Kirby.

**Visualization:** Otobo I. Ujah, Russell S. Kirby.

**Writing – original draft:** Otobo I. Ujah, Russell S. Kirby.

**Writing – review & editing:** Otobo I. Ujah, Jason L. Salemi, Rachel B. Rapkin, William M. Sappenfield, Elen M. Daley, Russell S. Kirby.

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
