## [Decision Letter · Decision Letter 0]

24 Nov 2023

PGPH-D-23-01986

Association of Missed Opportunity for Family Planning Counseling with Pregnancy Intention: A Multilevel Analysis of Data from the Performance Monitoring for Action (PMA) Surveys

Dear Dr. Ujah,

Thank you for submitting your manuscript to PLOS Global Public Health. After careful consideration, we feel that it has merit but does not fully meet PLOS Global Public Health’s publication criteria as it currently stands. Therefore, we invite you to submit a revised version of the manuscript that addresses the points raised during the review process.

Specifically, the reviewer raises a fundamental query regarding methodology as the definition of ‘missed opportunity for family planning’ does not consider multiple possible confounding factors, however these could be addressed by revisions or justified by further elaboration on your rationale. Please refer to the reviewers specific comments below and in the attached document.

We look forward to receiving your revised manuscript.

Kind regards,

Jennifer Tucker, PhD

Staff Editor

Journal Requirements:

Additional Editor Comments (if provided):

Please note that we have only been able to secure a single reviewer to assess your manuscript. We are issuing a decision on your manuscript at this point to prevent further delays in the evaluation of your manuscript. Please be aware that the editor who handles your revised manuscript might find it necessary to invite additional reviewers to assess this work once the revised manuscript is submitted. However, we will aim to proceed on the basis of this single review if possible. 

Reviewers' comments:

Reviewer's Responses to Questions

**Comments to the Author**

1. Does this manuscript meet PLOS Global Public Health’s publication criteria? Is the manuscript technically sound, and do the data support the conclusions? The manuscript must describe methodologically and ethically rigorous research with conclusions that are appropriately drawn based on the data presented.

Reviewer #1: Partly

2. Has the statistical analysis been performed appropriately and rigorously?

Reviewer #1: Yes

3. Have the authors made all data underlying the findings in their manuscript fully available (please refer to the Data Availability Statement at the start of the manuscript PDF file)?

Reviewer #1: Yes

4. Is the manuscript presented in an intelligible fashion and written in standard English?

Reviewer #1: Yes

5. Review Comments to the Author

Reviewer #1: My review is in the attached file. I have pasted in just the first few words to meet the character count.

This study undertakes a multilevel analysis of the association between the intendedness of a woman’s most recent pregnancy (in the past two years) and the likelihood of she experienced a missed opportunity for family planning (FP) counseling, using three cross-sectional rounds of PMA2020 survey data collected in Nigeria. The manuscript is well written, the methodology details are clear and properly interpreted. The research question warrants investigation. There are several measurement and confounding issues that need to be resolved.

6. PLOS authors have the option to publish the peer review history of their article (what does this mean?). If published, this will include your full peer review and any attached files.

**Do you want your identity to be public for this peer review?** For information about this choice, including consent withdrawal, please see our Privacy Policy.

Reviewer #1: No

---

## [Decision Letter · Decision Letter 1]

16 Apr 2024

PGPH-D-23-01986R1

Do Women with a Previous Unintended Birth Subsequently Experience Missed Opportunities for Postpartum Family Planning Counseling? A Multilevel Mixed Effects Analysis

Dear Dr. Ujah,

Thank you for submitting your manuscript to PLOS Global Public Health. After careful consideration, we feel that it has merit but requires few more minor edits to fully meet PLOS Global Public Health’s publication criteria. Therefore, we invite you to submit a revised version of the manuscript that addresses the points raised during the review process.

Both reviewers agree all prior comments have been adequately addressed and the manuscript is much improved now. But reviewer 1 has a few more comments I think are important to address. So please address the few pending comments before we can formally accept it for publication.

We look forward to receiving your revised manuscript.

Kind regards,

Patience A. Afulani, MBChB, MPH, PhD

Academic Editor

Journal Requirements:

Additional Editor Comments (if provided):

Reviewers' comments:

Reviewer's Responses to Questions

**Comments to the Author**

1. If the authors have adequately addressed your comments raised in a previous round of review and you feel that this manuscript is now acceptable for publication, you may indicate that here to bypass the “Comments to the Author” section, enter your conflict of interest statement in the “Confidential to Editor” section, and submit your "Accept" recommendation.

Reviewer #1: All comments have been addressed

Reviewer #2: All comments have been addressed

2. Does this manuscript meet PLOS Global Public Health’s publication criteria? Is the manuscript technically sound, and do the data support the conclusions? The manuscript must describe methodologically and ethically rigorous research with conclusions that are appropriately drawn based on the data presented.

Reviewer #1: Yes

Reviewer #2: Yes

3. Has the statistical analysis been performed appropriately and rigorously?

Reviewer #1: Yes

Reviewer #2: Yes

4. Have the authors made all data underlying the findings in their manuscript fully available (please refer to the Data Availability Statement at the start of the manuscript PDF file)?

Reviewer #1: Yes

Reviewer #2: Yes

5. Is the manuscript presented in an intelligible fashion and written in standard English?

Reviewer #1: Yes

Reviewer #2: Yes

6. Review Comments to the Author

Reviewer #1: My comments are in the attached file and pasted in below.

Thank you for the changes and improved manuscript. There are still some small concerns that need to be addressed but overall the analysis is on sounder ground.

1. Table 1 – I believe footnotes a and b need to be reversed.

2. Please clarify – on p. 18 the text refers to a sample size of 6,479 women and 822 clusters, while on p. 16 the number of clusters is 420. Tables 2 and 3 cite 822 clusters. Is 420 a typo? PMA2020’s sample design involved repeated use of clusters in the first 4 survey rounds (which were about 6 months apart), followed by a new selection of sample clusters for rounds 5 and 6 but that number would have been the same as those in rounds 1-4.

3. Please do a quick search for the use of “determinants” and replace with “factors associated with”. “Determinants” should be reserved for use with causal analyses.

4. On p. 17, the text (3 lines from bottom) says “nearly two thirds (58.5%)”; this should be “nearly three-fifths”, since 58.5% is closer to 60% than to 67%.

5. On p. 34, the narrative on limitations states, “Second, the ascertainment of MOs was based on women’s self-report.” This is not the case and was the point of my first comment, i.e., that a missed opportunity is operationally defined by the study researchers but is not self-reported by the female respondent. The woman was not directly asked if she experienced a missed opportunity for FP counseling. For example, she could have been asked if she was interested in receiving FP information after her delivery and, if so, where/when would have been the most appropriate time/place. From a health policy and service coverage perspective, all postpartum clients assessed by providers to have a contraceptive need should be offered services, and this is what is driving the researchers’ definition and study interests. Please adjust the wording here.

6. Last, just to underscore the importance of the cluster-level findings, I think these reflect a normative climate In Nigeria that does not support FP usage in general. Thus providers will not think to mention FP to PP clients or may not diligently follow MCH protocol to screen for past unplanned pregnancies. Mothers may be hesitant to ask about pregnancy prevention means in a public facility setting. I recommend the Discussion emphasize the need for further study of community attitudes and norms about FP that influence providers’ and mothers’ attitudes and comfort to receipt of such information after delivery.

Reviewer #2: This revision is well executed. The authors have handsomely addressed all the issues in the manuscript.

However, there are little issues with the introduction.

I think it will be more appropriate if the authors delete either the questions or the objectives. Both research questions or objectives indicate double standards in the manuscript. I suggest that the authors delete the research questions as the research objectives of the study are well-defined and written.

7. PLOS authors have the option to publish the peer review history of their article (what does this mean?). If published, this will include your full peer review and any attached files.

**Do you want your identity to be public for this peer review?** For information about this choice, including consent withdrawal, please see our Privacy Policy.

Reviewer #1: No

Reviewer #2: No

---

## [Decision Letter · Decision Letter 2]

10 May 2024

Do Women with a Previous Unintended Birth Subsequently Experience Missed Opportunities for Postpartum Family Planning Counseling? A Multilevel Mixed Effects Analysis

PGPH-D-23-01986R2

Dear Dr Ujah,

We are pleased to inform you that your manuscript 'Do Women with a Previous Unintended Birth Subsequently Experience Missed Opportunities for Postpartum Family Planning Counseling? A Multilevel Mixed Effects Analysis' has been provisionally accepted for publication in PLOS Global Public Health.

Best regards,

Julia Robinson

Executive Editor

Reviewer Comments (if any, and for reference):

Reviewer's Responses to Questions

**Comments to the Author**

1. If the authors have adequately addressed your comments raised in a previous round of review and you feel that this manuscript is now acceptable for publication, you may indicate that here to bypass the “Comments to the Author” section, enter your conflict of interest statement in the “Confidential to Editor” section, and submit your "Accept" recommendation.

Reviewer #1: All comments have been addressed

Reviewer #2: All comments have been addressed

2. Does this manuscript meet PLOS Global Public Health’s publication criteria? Is the manuscript technically sound, and do the data support the conclusions? The manuscript must describe methodologically and ethically rigorous research with conclusions that are appropriately drawn based on the data presented.

Reviewer #1: Yes

Reviewer #2: Yes

3. Has the statistical analysis been performed appropriately and rigorously?

Reviewer #1: Yes

Reviewer #2: Yes

4. Have the authors made all data underlying the findings in their manuscript fully available (please refer to the Data Availability Statement at the start of the manuscript PDF file)?

Reviewer #1: Yes

Reviewer #2: Yes

5. Is the manuscript presented in an intelligible fashion and written in standard English?

Reviewer #1: Yes

Reviewer #2: Yes

6. Review Comments to the Author

Reviewer #1: All my prior comments have been addressed.

Reviewer #2: The authors have addressed all the necessary concerns. I think the manuscript is eligible for publication. Thank you

7. PLOS authors have the option to publish the peer review history of their article (what does this mean?). If published, this will include your full peer review and any attached files.

**Do you want your identity to be public for this peer review?** For information about this choice, including consent withdrawal, please see our Privacy Policy.

Reviewer #1: No

Reviewer #2: No
